# Watershed zonation through hillslope clustering for tractably quantifying above- and belowground watershed heterogeneity and functions

Haruko M. Wainwright[1,2], Sebastian Uhlemann[1], Maya Franklin[1], Nicola Falco[1], Nicholas J. Bouskill[1], Michelle E. Newcomer[1], Baptiste Dafflon[1], Erica Woodburn[1], Burke J. Minsley[3], Kenneth H. Williams[1,4], Susan S. Hubbard[1]

[1]Lawrence Berkeley National Laboratory, Berkeley, CA 94720, USA
[2]University of California, Berkeley, CA 94720, USA
[3]U.S. Geological Survey, Denver, CO 80225, USA
[4]Rocky Mountain Biological Laboratory, Crested Butte, CO, 81224, USA

*Correspondence to*: Haruko M. Wainwright (hmwainwright@lbl.gov)

**Abstract.** In this study, we develop a watershed zonation approach for characterizing watershed organization and functions in a tractable manner by integrating multiple spatial data layers. We hypothesize that (1) a hillslope is an appropriate unit for capturing the watershed-scale heterogeneity of key bedrock-through-canopy properties, and for quantifying the co-variability of these properties representing coupled ecohydrological and biogeochemical interactions; (2) remote sensing data layers and clustering methods can be used to identify watershed hillslope zones having the unique distributions of these properties relative to neighbouring parcels; and (3) property suites associated with the identified zones can be used to understand zone-based functions, such as response to early snowmelt or drought, and solute exports to the river. We demonstrate this concept using unsupervised clustering methods that synthesize airborne remote sensing data (LiDAR, hyperspectral, and electromagnetic surveys) along with satellite and streamflow data collected in the East River Watershed, Crested Butte, Colorado, USA. Results show that (1) we can define the scale of hillslopes at which the hillslope-averaged metrics can capture the majority of the overall variability in key properties (such as elevation, net potential annual radiation and peak SWE), (2) elevation and aspect are independent controls on plant and snow signatures, (3) near-surface bedrock electrical resistivity (top 20 m) and geological structures are significantly correlated with surface topography and plan species distribution, and (4) K-means, hierarchical clustering, and Gaussian mixture clustering methods generate similar zonation patterns across the watershed. Using independently collected data, we show that the identified zones provide information about zone-based watershed functions, including foresummer drought sensitivity and river nitrogen exports. The approach is expected to be applicable to other sites and generally useful for guiding the selection of hillslope-experiment locations and informing model parameterization.

## 1 Introduction

Predictive understanding of watershed functions is often hindered by the heterogeneous and multiscale fabric of watersheds (e.g., Peters-Lidard et al., 2017). Heterogeneity exists within each of the watershed compartments, including aboveground compartments (i.e., plant species distribution and plant dynamics, topography) and belowground compartments (i.e., soil and bedrock structures/properties). Such watershed *patterns* influence ecohydrological and biogeochemical *processes*, which in turn affect watershed *functions* and create emerging *patterns* as feedback (Sivapalan, 2006). Since watersheds consist of diverse biotic and abiotic compartments, watershed functions include diverse signatures, including hydrological (i.e., partition, storage, and release of water), ecological (e.g., species adaptation, productivity), and geochemical (e.g., nutrient cycling, solute export) signatures (Sivapalan, 2006; Wagener et al., 2007). There is a multiscale nature of heterogeneity such that different processes have different characteristic scales. Watershed hydrology modeling studies typically use the grid size from 100 m to 1 km (e.g., Foster et al., 2020; Maina et al., 2020 ), whereas soil moisture is known to vary on the order of several or several tens of meters (e.g., Engstrom et al., 2005; Wainwright et al., 2015), and biogeochemical dynamics can vary within one meter or less (e.g., Burt and Pinay, 2005; Groffman et al., 2009).

There have been extensive studies investigating how the heterogeneous watershed organization influences water, energy, and nutrient cycling and their fluxes (e.g., Peters-Lidard et al., 2017). There are two directions for tackling this problem: a bottom-up Newtonian approach or a top-down Darwinian approach. The Newtonian approach is a reductionist approach that describes a system by a set of mass/energy/momentum conservation equations with spatially variable parameters. Recently, integrated hydrological and reactive transport models have been successfully implemented to describe and predict watershed behaviors from hillslope to watershed scales (e.g., Maxwell and Kollet, 2008; Li et al., 2017). In addition, the Hydrological Response Unit concept has been used to classify the landscapes based on spatial datasets (e.g., landcover types, elevation, and soil maps) and to parameterize hydrological models (Flugel, 1997; Aytaç, 2020). On the other hand, the Darwinian approach identifies rules or organizing principles governing spatial patterns of complex datasets and defines watersheds as self-organized and co-evolved units by watershed functional traits (McDonnell et al., 2007). Catchment scaling and similarity concepts have been used to synthesize the catchment datasets across scales and to classify catchments (Wagener et al., 2007; Thompson et al., 2011; Sawicz et al., 2011; Krause et al. 2014).

In parallel, there have been recently significant advances in understanding and quantifying the watershed-scale heterogeneity of bedrock-to-canopy terrestrial compartments, which regulate water and nutrient cycling and their exports; particularly through the critical zone observatory (CZO) network (Brantley et al., 2017). In particular, Pelletier et al. (2018) have highlighted the control of slope aspects on ecosystem and critical-zone systems, finding that, for example, in water-limited systems, the north-facing slopes have less evapotranspiration and hence higher soil moisture, deeper weathering, and larger nutrient retention in soil (e.g., Hinckley et al., 2014). Such advances are largely attributed to a variety of spatially extensive

characterization technologies across bedrock-to-canopy compartments, which provide various *patterns*. High-resolution DEM from LiDAR have been applied to better understand the relationship between geomorphology and hydrology (Prancevic and

65 Kirchner, 2019), as well as to measure snow depths and snow-water-equivalent (SWE) over a basin scale (Painter et al., 2016). LiDAR data was also able to inform near-surface soil properties (Patton et al., 2018; Gillin et al., 2015), hydrological connectivity (Jensco et al., 2009), and biogeochemical hotspots (Duncan et al., 2013). In addition, hyperspectral remote sensing can map plant traits (e.g., Asner et al., 2015), leaf water content (e.g., Colombo et al., 2008), leaf chemistry (e.g., Feilhauer, et al., 2015) and other properties, which are also proxies for soil biogeochemistry (Madritch et al., 2014). At the same time,

geophysics has been extensively used to characterize the subsurface structure and to estimate soil and bedrock properties. Surface geophysics has been used to measure bedrock depth, weathering zone thickness, and other properties (e.g., de Pasquale et al., 2019), contributing to hillslope-scale hydrological characterization and modeling. Surface ERT and seismic data have also revealed the influence of tectonic stresses and hydrological processes on bedrock fracturing and weathering (Rempe and Dietrich, 2014; St. Clair et al., 2015). Airborne geophysics—particularly airborne electromagnetic (EM) surveys—was

originally developed for mineral exploration, but is now increasingly used for water-resources applications (e.g., Barfod et al., 2018; Ball et al., 2020).

Despite these advancements, there are still challenges in associating watershed *functions* to heterogenous watershed *patterns*. It remains a challenge to integrate multitype and multiscale datasets, including ground-based point measurements and airborne

or satellite remote-sensing datasets. Although the aboveground properties, such as topography and plant characteristics, can be mapped over the watershed scale, the subsurface variability is still difficult to map over that scale, which is one of the biggest uncertainties in hydrology (Fan et al., 2019). In addition, even though hillslope-scale characterization and experiments (such as tracer tests) can be extremely useful for providing detailed information about watershed functions (e.g., Hinkley et al, 2014), it is difficult to select several hillslopes for such intensive characterization, or to gauge the representativeness of one

hillslope for an entire watershed.

In this study, we develop and test the concept of a zonation approach for tractably characterizing the organization of a watershed based on multiple spatial data layers, and how these characteristic *patterns* aggregate to predict watershed *functions*. The clustering-based zonation approaches have emerged recently as effective spatial data integration methods that use spatial

clustering to identify regions or zones that have unique distributions of heterogeneous properties and key functions relative to neighboring regions (Hubbard et al., 2013; Wainwright et al., 2015; Devadoss et al., 2020; Hermes et al., 2020). For watershed zonation, we consider a hillslope as a fundamental unit for watershed hydrology and element cycling, funneling water and elements from the ridge to the river (Fan et al., 2019), and also representing aspect controls on critical zones (Pelletier et al., 2018). We follow Band (1989) and Band et al. (1991; 1993) to investigate the appropriate scales of hillslopes for capturing the

watershed heterogeneity, while limiting the internal variance within hillslopes.

We then hypothesize that (1) a hillslope is an appropriate unit for capturing the watershed-scale heterogeneity of key bedrock-through-canopy properties, and for quantifying the co-variability of these properties representing coupled ecohydrological and biogeochemical interactions, (2) we can identify a group of hillslopes or watershed-scale zones that have unique distributions of these properties relative to neighboring parcels, and (3) the identified zones can capture the variability of key watershed functions. We demonstrate our approach using the airborne and spatial datasets collected in the East River watershed region near Crested Butte, Colorado, USA (Hubbard et al., 2018). We apply and compare multiple clustering methods to understand the characteristics, commonality, and differences among each method. Finally, we validate the zonation hypothesis based on the datasets that define key watershed functions, including drought sensitivity of plant productivity and water/nitrogen export.

## 2 Site and Data

We consider the domain of approximately 15 km by 15 km (Figure 1) near Gothic, Colorado, USA, which is the same area used in a recent study (Wainwright et al., 2020). As described in Hubbard et al. (2018), the domain includes four catchments, including the East River, Washington Gulch, Slate River and Coal Creek. It is a part of the Elk Range in the Rocky Mountains, with elevation from 2800 m to 4000 m (Figure 1a). The major land-cover types (NLCD 2011) are rock outcrop (12%), evergreen forest (29%), deciduous forest (18%), grassland (30%), and woody wetland (6%). Geology within the domain is diverse, including Paleozoic, Mesozoic, and Cenozoic sedimentary rocks (siltstones and sandstones of the late Permian Maroon Formation; shales and sandstones of the upper Cretaceous Mancos Shale and Mesaverde group, respectively; siltstones and sandstones of the Eocene Wasatch Formation) and Miocene igneous intrusive rocks of predominantly granodioritic composition (Gaskill et al., 1991).

The spatial data layers include the USGS landcover map (NLCD 2011), the digital geological map of Colorado (Green et al., 1992), and the soil texture maps (%clay and %sand) from the POLARIS database (Chaney et al., 2016; 2019). In addition, we used four airborne datasets (Texts S1): an airborne electromagnetic (AEM) survey acquired in fall 2017 (Minsley and Ball, 2018; Uhlemann et al., submitted; Zamudio et al., 2020), LiDAR and hyperspectral data collected by the National Ecological Observation Network (NEON) team in June 2018 (Chadwick et al., 2020), and NASA Airborne Snow Observatory (ASO) data collected in April 2018 (doi.org/10.5067/M4TUH28NHL4Z).

To test the zonation hypothesis, we used datasets representing two key functions: foresummer drought sensitivity of plant productivity and river nitrogen export. The foresummer drought sensitivity map based on Landsat Normalized Difference Vegetation Index (NDVI) was developed by Wainwright et al. (2020) to represent the plant productivity responses to early snowmelt and subsequent drought conditions in the primary growing season. To create this map, Wainwright et al. (2020) performed the linear regression of the historical peak Landsat NDVI as a function of the Palmer Drought Severity Index in June. They then defined the slope of the linear fit as the foresummer drought sensitivity, which represented the magnitude of

peak plant productivity changes with respect to the drought condition in the growing season. Since the satellite images were
not used in clustering, they were considered independent datasets. In addition, the annual discharge and nitrogen export were
computed from the streamflow and chemistry data at the subcatchments (Figure 1) within the domain used in Carroll et al.
(2018). The nitrogen export was computed in the same manner as Newcomer et al. (2021).

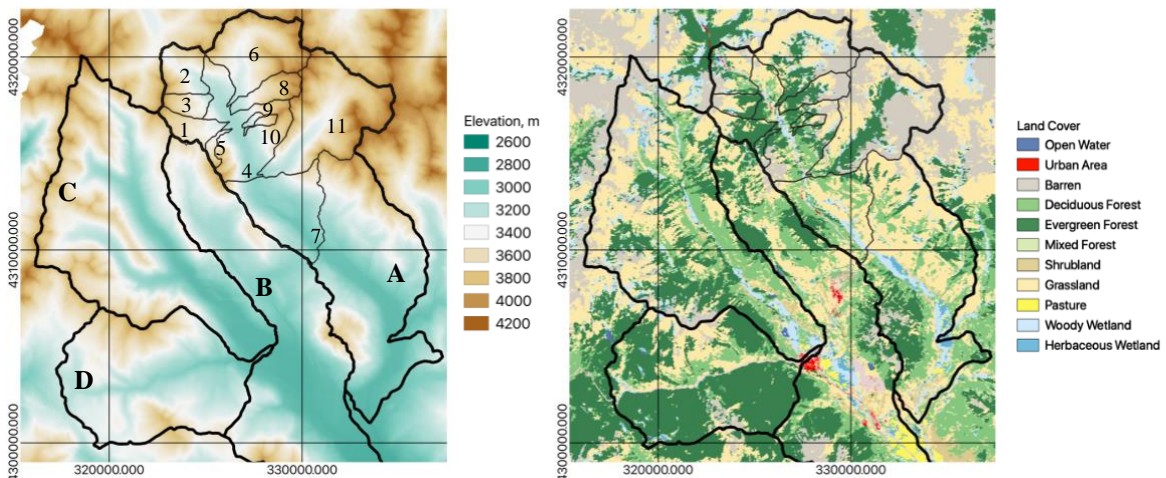

**Figure 1. Study domain with (a) elevation and (b) land cover map (USGS NLCD 2011). The black lines are the boundaries of the four watersheds: (A) East River, (B) Washington Gulch, (C) Slate River and (D) Coal Creek. The thin black boundaries are subcatchments of the East River watershed from Carroll et al. (2018): (1) Rock, (2) East Above Quigley (EAQ), (3) Quigley, (4) East Below Copper (EBC), (5) Gothic, (6) Rustlers, (7) Pumphouse, (8) Bradley, (9) Marmot, (10) Avery, and (11) Copper.**

## 3 Methodology

### 3.1 Watershed and Hillslope Characteristics

We followed Wainwright et al. (2020) to compute pixel-by-pixel topographic metrics based on the LiDAR Digital Elevation Model (DEM), including slope, topographic wetness index (TWI), and annual net potential solar radiation. We selected metrics based on the previous studies that reported their relevance to soil moisture, soil thickness, water quality and others (Mohanty
et al., 2000; Gillin et al., 2015; Lintern et al., 2018). The annual net potential solar radiation – a function of the aspect, slope, and solar angle – is considered as a better metric to represent the intensity of solar radiation than the aspect itself which is circular (0 and 360 degrees are the same; Wainwright et al., 2020). In parallel, we delineated each hillslope based on the DEM, following Noel et al. (2014) and using Topotoolbox (Schwanghart, 2014). The hillslope delineation algorithm first identifies the stream segments that are a collection of pixels that have larger flow accumulation (i.e., a larger drainage area) than a certain

threshold value, using the flow routing algorithms in Topotoolbox. It then finds the pixels in both sides of each stream segment, and also the drainage area leading to each of these pixels. This process yields two lateral hillslopes in both sides of each stream segment. For the first-order stream, the algorithm also defines the headwater hillslope, which is the drainage area leading to the pixel at the origin of the stream. Although these hillslopes are based on surficial water routing, we assume that the DEM captures near-surface hydrological connectivity as documented by Jensco et al. (2009).


We then defined the characteristics of each hillslope, based on the spatial data layers (Figure S1). We computed the average values of the DEM topographic metrics, AEM-based bedrock resistivity, NEON products (Normalized Difference Vegetation Index, NDVI; Normalized Difference Water Index, NDWI, and biomass) at the peak growing season (2018), peak SWE (2018) and soil texture in each hillslope. In addition, we computed the relief of each hillslope, which was the difference between the
minimum and maximum elevation. For the categorical variables such as land-cover types and geology, we computed the percent coverage of each plant type and surface geology in each hillslope. These 17 hillslope features are defined for each hillslope (Table S1).

As Band (1989) and Band et al (1991) noted, the hillslope delineation can create different sets of hillslopes, depending on the
threshold drainage areas that define the stream segments. For the hillslope metrics defined by the average (e.g., elevation, slope, and others), we evaluated how these averaged metrics can represent the overall variability of the watershed properties, and how this representation changes depending on the different threshold drainage areas. We consider that the variance of pixel-by-pixel properties represents the overall variability over this domain, while the variance of hillslope-averaged metrics represents the variability across the hillslopes. We computed the ratio between the across-hillslope variance and the overall
variance, representing how much watershed-scale variability the hillslopes can capture. In addition, as a contrast, we computed the variance of each property within the upscaled pixels, by taking the averaged values in larger pixels compared to the original 9-meter pixel. We computed the ratio between this across-upscaled-pixel variance and the overall variance. In this way, we can investigate the difference between hillslope-based spatial aggregation compared to standard pixel-based upscaling for each of the key watershed properties.


### 3.2 Cluster-Based Approach to Identify Watershed Zones

Based on the hillslope features, we first evaluated the correlations among multivariate above/belowground properties. Although such multivariate co-variability has been analyzed using principal component analysis (Devadoss et al., 2020), we used scatter plots and correlation coefficients in this study because of nonlinearity. We then applied three commonly used
unsupervised clustering methods: K-means (KM), hierarchical clustering (HC), and Gaussian mixture models (GMM; Hastie et al., 2001; Kassambara, 2017). We scaled each feature by the mean and standard deviation, and defined the dissimilarity between two data points based on the Euclidean distance. The characteristics of each method are described in Texts S2.

Multiple methods are often evaluated based on true classes or labeled datasets (Rodriguez et al., 2019), which are not available here. We used a silhouette score, which represented how similar a given data point was to its own cluster compared to other clusters. For GMM, we used the Bayesian Information Criterion (BIC) for selecting the number of clusters.

After the clusters are defined, we transfer the clusters—the group of hillslopes that have similar features—to the spatial map as zones. We identify the common zones across the three methods as well as the zones that differ. We then evaluate the distribution of hillslope features and functions in each zone using box plots to define the characteristics of each zone. The foresummer drought sensitivity (Wainwright et al., 2020) is a watershed function available throughout the domain that allows us to quantify the hillslope-average values in the same manner as other spatial data layers. In addition, we computed the spatial coverage of each zone in each subcatchment, and compared them to the ratio between the annual nitrogen export and total discharge, which is considered as a key metric indicating how watersheds retain and lose nutrients (Newcomer et al., 2021).

## 4. Results

### 4.1 Hillslope Scales

The ratio between the across-hillslope variance and the overall variance (Figure 2a) is generally high for the elevation, net annual potential solar radiation (radiation), peak SWE and bedrock resistivity up to 0.75, which means that the hillslope averaged metrics capture the watershed-scale variability of these variables and that the within-hillslope variability is small compared to the across-hillslope variability. TWI and NDVI, on the other hand, have a low ratio, which means that the within-hillslope variability is significant. The ratio increases as the drainage area decreases, since the smaller the hillslopes, the better they capture small-scale variability. However, the variance ratio of the elevation and radiation reaches a plateau with the drainage area around $10^6 m^2$. This means that the internal variability within hillslopes is limited up to this threshold drainage area, and that the hillslope metrics are representative of the overall variability up to this threshold. Based on this result, we selected 810,000 $m^2$ as the threshold drainage area in the subsequent analysis.

We can compare this hillslope-based averaging (Figure 2a) with pixel-based averaging/upscaling (Figure 2b). Similar to hillslope-based averaging, the ratio between the across-upscaled-pixel variance and the overall variance increases as the pixel size decreases. The magnitude is similar such that the elevation, peak SWE and bedrock resistivity have a higher ratio, meaning that the upscaled properties can capture the overall variability of these properties. The exception is the radiation, since the pixel-averaged radiation captures only up to 60% of the overall variance, while the hillslope-averaged radiation captures up to 70%. In addition, different from the hillslope average, the ratio for radiation in Figure 2b keeps increasing without reaching a plateau. This means that a representative size or scale does not exist when we use pixel-based upscaling.

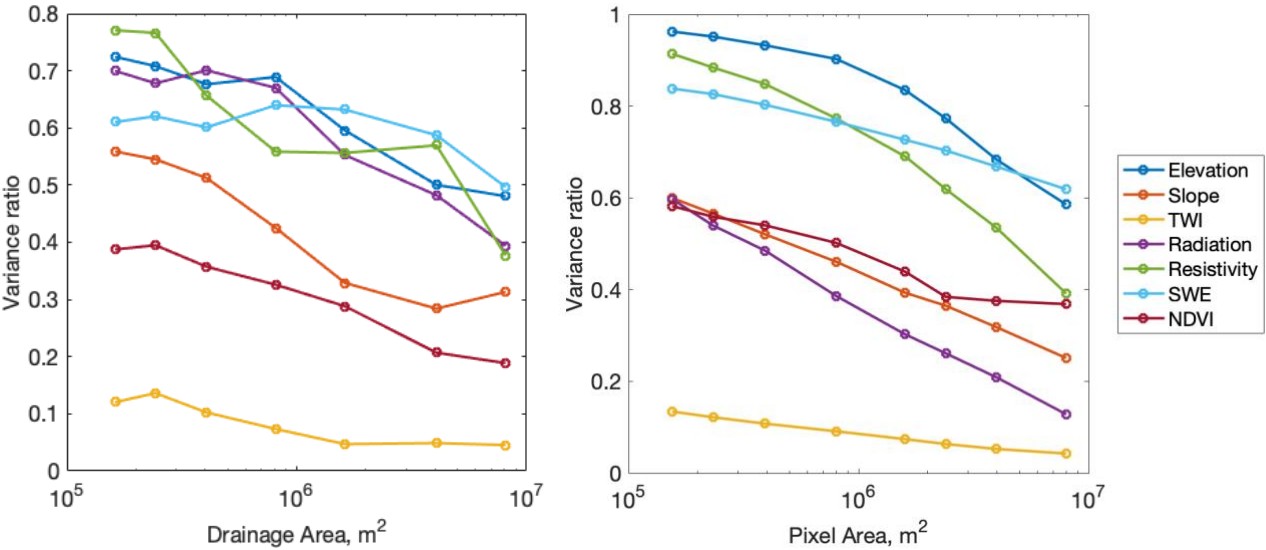

Figure 2. Variance ratio (a) between the across-hillslope variance and the overall variance as a function of the threshold drainage area, and (b) between the across-upscaled-pixel variance and the overall variance as a function of the pixel area.

## 4.2 Watershed Zonation

Figure 3 shows that the above- and belowground hillslope features are significantly correlated with each other. In particular, elevation is correlated with all other features except for net annual potential radiation. The hillslopes at higher elevation have steeper slopes and lower TWI, higher bedrock resistivity and higher peak SWE. There are some nonlinearities: TWI increases beyond the linear function at lower elevations. The relationship between peak NDVI and elevation is quadratic, having peaks in mid-elevation (corresponding to around 3300 m). The net annual potential radiation is weakly correlated with slope, TWI, and peak SWE. The correlation coefficients among the hillslope-averaged features are significantly higher than the pixel-by-pixel ones (Figure S2).

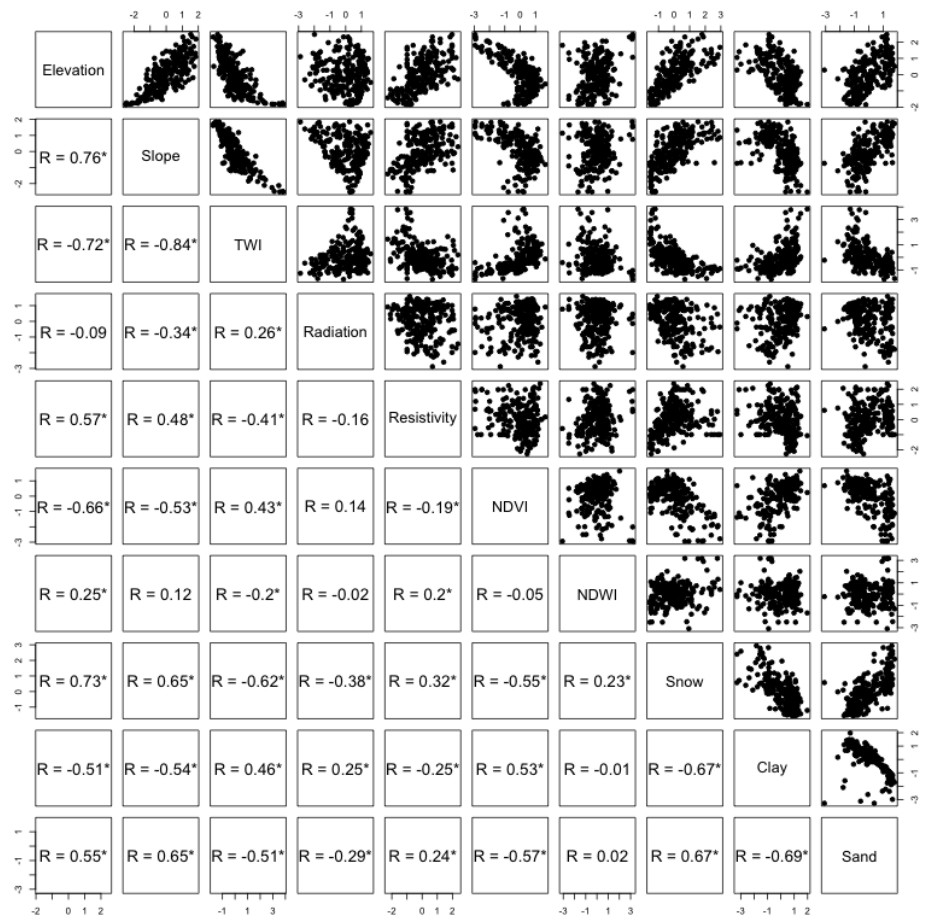

**Figure 3. Correlation and scatter plots (Pearson's correlation coefficients) among selected hillslope features (Table S1). The * sign represents p-values < 0.01.**

The clustering results are then mapped spatially as the watershed zones (Figure 4). The number of clusters is six, since the highest silhouette score is high at six clusters across the three methods (Figure S3a), and BIC for GMM is also the highest at six clusters (above four; Figure S3b). To compare the results from these three methods, we first identified six common zones across the three methods that have the overlapping coverage, starting from the GMM-based map as a basis. Zone 7 appears only in the HC result (Figure 4c) and is designated as a separate zone; the hillslopes in Zone 7 are included either in Zone 3 or

4 in the GMM and KM results. The frequency maps (Figure S4) show that most hillslopes are consistently categorized in the same zones among the three methods. The zonation maps from GMM (Figure 4a) and KM (Figure 4b) are quite similar, although small fractions of the hillslopes have different designations between the two maps. The zonation map from HC (Figure 4c) is also similar to the ones from GMM and KM except for Zone 7 which is a part of Zone 3 or 4 in the GMM and KM result.

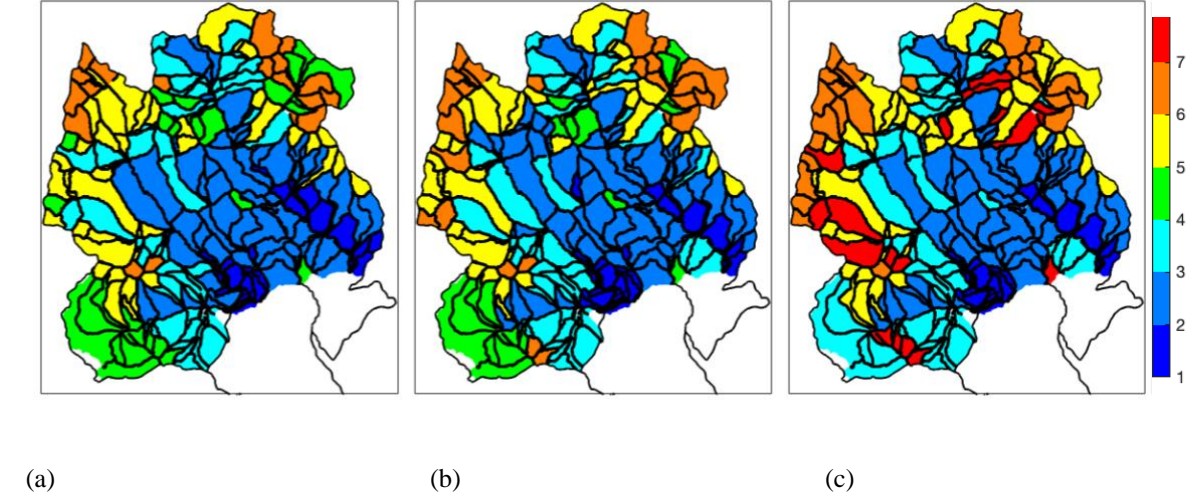


(a)                                    (b)                                    (c)

**Figure 4. Hillslope zonation based on (a) Gaussian mixture model (GMM), (b) K-means (KM), and (c) hierarchical clustering (HC). Different zones are represented by the cluster index from 1 to 7. There are 6 common zones among the three methods, and 1 additional zone (Zone 7) appears in the HC result (c). The black lines are the hillslope boundaries.**


Box plots shows the characteristics of each identified zone (Figures 5 and 6). Although each map has six zones, we use seven zones that are found across the three clustering methods. The statistics are computed by aggregating the zonation from the three methods (Figure S4) such that the zonation of each method has the weight of 1/3. Elevation (Figure 5a) increases from Zone 1 to 6, while Zone 7 is similar to Zone 4. Slope (Figure 5b) shows a similar progression to elevation, which is consistent with the correlation plot (Figure 3), although Zone 7 has steeper hillslopes than expected for its elevation range. Relief (Figure 5c) also has a similar order to the elevation, except that Zones 4 to 7 are similar, because the higher elevation hillslopes are smaller with lower relief, even though the slope is high. TWI (Figure 5d) has an opposite trend to elevation, consistent with the correlation plot (Figure 3). In terms of the net annual potential radiation (Figure 5e), the grassland-dominated hillslopes (Zones 1, 2 and 5) tend to be higher, while the conifer-dominated hillslopes (Zones 3, 4 and 7) are lower except for Zone 4 being around the average. The bedrock resistivity (Figure 5f) is higher at the higher elevation zones (Zones 4 – 7) or an intrusive-rock-dominated zone (Zones 4). NDVI (Figure 5g) is higher in the forest-dominated zones (Zone 2, 3, 4, and 7). Zone 6 has significantly lower NDVI, since it is predominantly in the barren region. NDWI (Figure 5h) is lower in the low elevation zones (Zones 1). Peak SWE (Figure 5i) is associated with the elevation such that it is the highest in Zone 6, which has the highest elevation, and the lower net potential radiation.

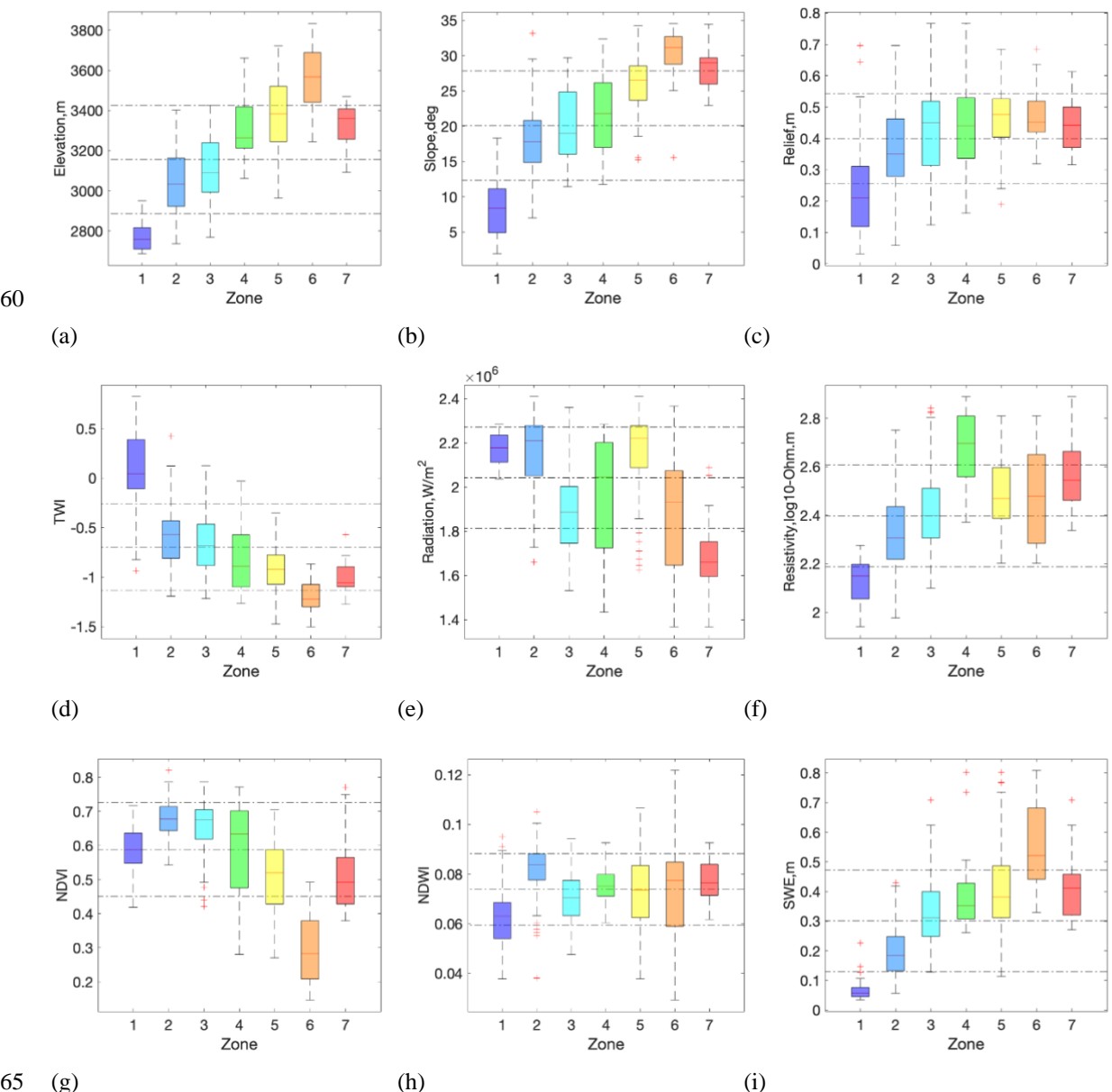

(a)                            (b)                            (c)

(d)                            (e)                            (f)

(g)                            (h)                            (i)

**Figure 5. Boxplots of hillslope pattern features: (a) elevation, (b) slope, (c) relief, (d) TWI, (e) annual net potential radiation, (f) bedrock electrical resistivity (top 20 m), (g) peak NDVI in 2018, (h) NDWI (at peak NDVI) in 2018, and (i) peak SWE in 2018. The horizontal lines are the overall mean and plus/minus one standard deviation.**

With respect to geology (Figure 6a-c), the low-elevation zones (Zone 1 and 2) are predominantly shale, while the mid-to-high elevation zones (Zone 3 and 5) have both shale and sandstone. Zone 4, 6 and 7 have a high percentage of intrusive rock coverage. Among the conifer-dominated areas (Zones 3, 4, and 7), Zone 4 is associated with intrusive rock, while Zone 3 is

associated with shale and sandstone, and Zone 7 has the mixture of these three bedrock types. In terms of land-cover types (Figure 6d-g), the barren area coverage (Figure 6d) increases with elevation from Zone 1 to 6. The grassland region (Figure

6e) exists across all zones, but it is particularly high in the low-elevation zones (Zones 1 and 2) and the higher-elevation south-facing zone (Zone 5). The coverage of deciduous forests (Figure 6f) is higher at lower elevation (i.e., Zones 1 and 2); Zone 2, in particular, has high aspen coverage. The evergreen forest regions (Figure 6g) appear in Zones 3, 4, and 7, which are associated with different bedrock types and slopes as mentioned above. Finally, the wetland (woody and herbaceous combined) coverage is higher in the low-elevation Zone 1. The characteristics of each zone are summarized in Table 1.


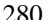


**Figure 6. Boxplots of hillslope pattern features with respect to geology and land-cover types: the percent coverage of (a) shale, (b) sandstone, (c) igneous (intrusive) rock, (d) barren, (e) grassland, (f) deciduous forest, (g) evergreen forest and (h) riparian zone.**

**Table 1. Summary of the features in each zone based on Figures 5 and 6.**

| Zone | Elevation | Slope | TWI | Resistivity | Radiation | Peak NDVI | Peak SWE | Plant types | Geology |
|---|---|---|---|---|---|---|---|---|---|
| 1 | Low | Low | High | Low | Mid-High | Mid | Low | Grassland | Shale |
| 2 | Low-Mid | Low-Mid | Mid-High | Low-Mid | Mid-High | Mid-High | Low-Mid | Grassland-Deciduous | Shale |
| 3 | Mid | Mid | Mid-High | Mid | Low | Mid-High | Mid | Evergreen | Shale-Sandstone |
| 4 | Mid | Mid-High | Low-Mid | High | Mid | Mid-High | Mid-High | Evergreen | Intrusive |
| 5 | Mid-High | Mid-High | Low-Mid | Mid-High | Mid-High | Low-Mid | Mid-High | Grassland | Shale-Sandstone |
| 6 | High | High | Low | High | Mid-Low | Low | High | Barren | Shale-Intrusive |
| 7 | Mid-High | High | Low-Mid | Mid-High | Low | Low-Mid | Mid-High | Evergreen | Mixed |

## 4.3 Watershed Functions

Foresummer drought sensitivity (Figure 7a) has an opposite trend to elevation (Figure 5a), with higher sensitivity in lower-elevation zones (Zone 1 and 2), and the lowest in Zone 6. Zone 5 is an exception such that this zone – high-elevation south-
facing grassland-dominated hillslopes – has higher sensitivity than the zones that are in a similar elevation range. The conifer-dominated zones (Zones 3, 4 and 7) have lower-than-average sensitivity. Tukey's pairwise comparison metrics suggest that the differences are significant except in the case of conifer-dominated Zones (3, 4 and 7).

We found that the magnitude of annual nitrogen exports scaled log-linearly with discharge, albeit with bifurcation among some
of the smaller catchments (Figure 7c). The Rock and Gothic subcatchments, which are predominantly within conifer-dominated Zone 3, 4 and 7 (Figure 7b), have a lower nitrogen expected from this simple scaling relationship. By contrast, Zone 2-dominated subcatchments (Marmot, Avery) have a larger nitrogen export than would be expected from this relationship. In addition, we evaluated the association of the N mass export and % coverage of each zone quantitatively. Since the number of points is not large, we divided the N export data into the two groups of the subcatchments that have the coverage
of each zone larger than 50% or less (Figure 7d and e). The association is statistically significant for Zone 2 and for the conifer-dominated zones combined (Zone 3, 4 and 7) with the *p*-value of less than 0.01. The larger coverage of Zone 2 is associated with a higher N export, while the conifer-dominated zones are associated with a lower export. The individual data points are also shown in Figure S5.

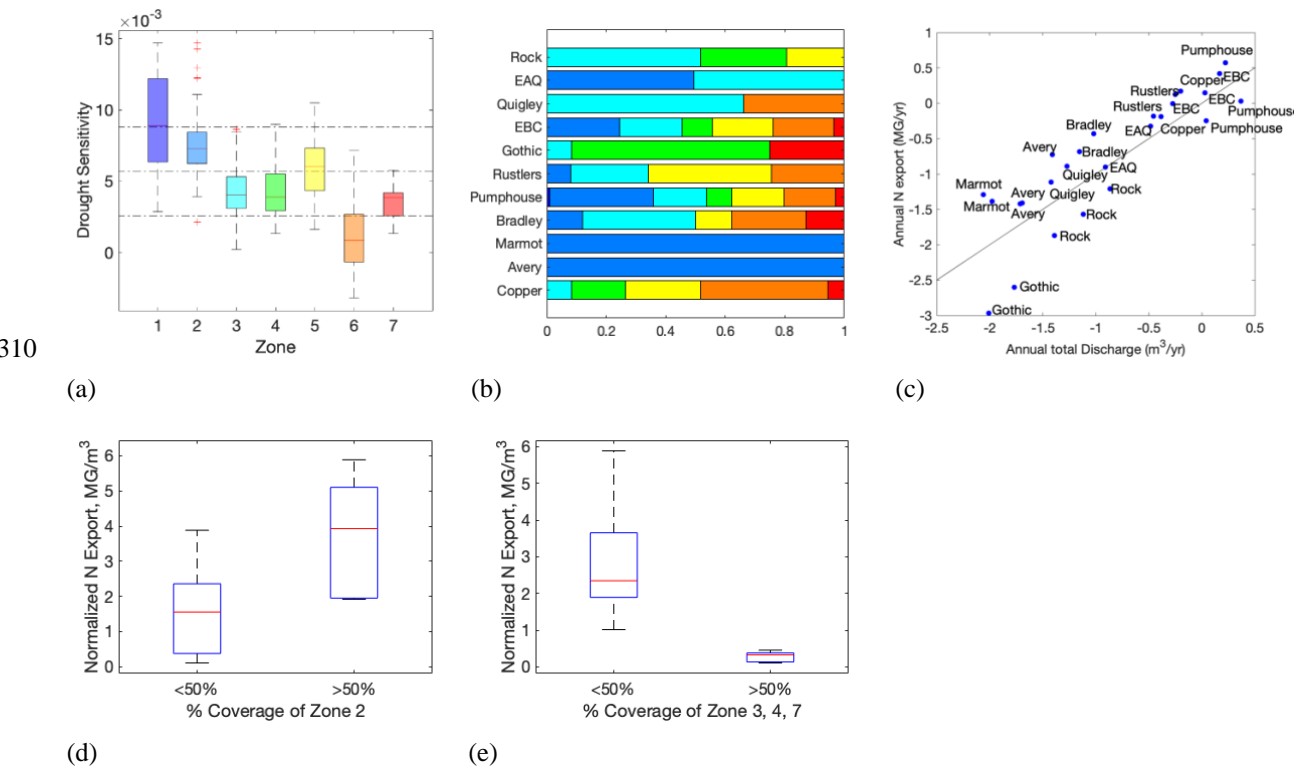


(a)                        (b)                        (c)

(d)                        (e)

**Figure 7. (a) Boxplot of the foresummer drought sensitivity as a function of zones, (b) zonation coverage of each subcatchment, (c)**
**annual discharge vs N export in 2015, 2016 and 2017, and (d, e) normalized N export (N export divided by annual discharge) as a**
**function of the percent coverage of (d) Zone 2 and (e) Zone 3, 4 and 7 as the conifer-dominated zone, respectively. In (b), the color**
**code is the same as Figures 3 – 5.**

## 5 Discussion

We first identified the co-variability between topographic metrics, elevation, bedrock resistivity, and plant signatures at the
watershed scale. In particular, the near-surface bedrock electrical resistivity is significantly correlated to slope and elevation.
The detailed analysis of AEM by Uhlemann et al. (submitted to Scientific Advances) has found that, (1) bedrock resistivity is
primarily controlled by bedrock type, and (2) within the shale and sandstone, the bedrock resistivity is affected by the extent
of hydrothermal alteration. The relationship between topography, elevation, and bedrock strength has been documented
extensively (e.g., Selby, 1987; Clarke and Burbank, 2010). Montgomery (2001) showed that, in low-precipitation and low soil
production environments, such as the East River, the erosion rate is dependent on rock strength and slope steepness, whereby
erosion-resistant rock (i.e., hydrothermally altered sandstones and shales and/or igneous rocks) are on steeper slopes than more
erodible rocks. Erosion-resistant rocks also remain at higher elevations, while erodible rock erodes away. This work provides

information on how surface topographic features are linked to bedrock variability and could be potentially used to inform bedrock variability, which is considered to be one of the largest uncertainties in watershed science (Fan et al., 2019).


In addition, our results show that plant signatures and peak SWE are primarily correlated with elevation. The quadratic dependency of NDVI and aboveground biomass on elevation is consistent with previous studies (e.g., Wainwright et al., 2020); productivity and biomass increases with elevation up to 3300-3500 m because plants have more access to snow-derived water at higher elevation. Above 3500 m, plant productivity is limited by temperature. Net annual potential radiation is not correlated

with elevation, but associated with conifer coverage and peak SWE. The conifer forest and high vegetative biomass are associated with north-facing slopes due to higher water availability, which is consistent with previous syntheses studies (e.g., Pelletier et al. 2018). The spatial variability of peak SWE is also consistent with previous studies being associated with elevation and radiation (Anderson et al., 2016). In addition, the conifer-dominated zones are associated with high-resistivity bedrock, while aspen forests are located within the less resistive bedrock. This is consistent with past studies showing that

conifer trees grow in the shallow bedrock regions, while aspens require the establishment of deeper root systems (e.g., Burke and Kasahara, 2011). Although there have been studies that have reported the co-variability of plants, topographic metrics, and aspect-controlled bedrock weathering (e.g., Pelletier et al, 2018), our analysis suggested an additional control of geological variability.

To capture such co-variabilities, we hypothesized that hillslopes would be appropriate spatial units to aggregate key properties and to classify into zones. However, the hillslopes could be defined at different scales based on the threshold drainage area, as documented by Band (1989) and Band et al. (1991). We computed the variance in hillslope-averaged metrics compared to the overall variance as a measure for quantifying how well hillslopes could capture the overall heterogeneity. We then investigated the effect of the threshold drainage area on this variance ratio. Our results show that, although this ratio decreases as the

drainage area increases, it stays flat up to a certain threshold drainage area, particularly for elevation and annual net potential radiation. This suggests that there is a representative scale of hillslopes, up to which the within-hillslope variability is limited, and the hillslope-averaged metrics can capture most of the overall variability across the domain. Such a representative scale has been known to exist as a representative elementary area (Wood et al., 1988). This representative scale is important, since Pelletier et al. (2019) showed the significant impact of hillslope aspects and slopes on evapotranspiration, weathering, nutrient

cycling, and others.

On the other hand, standard pixel-based upscaling (i.e., taking an average in larger pixels) does not show such a plateau, since larger pixels include a mixture of hillslopes with different aspects. This is consistent with Band et al. (1991) which documented that the hillslope-based partition had much greater efficiency for upscaling than pixel-based scaling by minimizing the within-

unit variability and maximizing the ability to capture the watershed-scale variability. Although pixel-based (or grid-based) spatial scaling is still the mainstream for remote sensing and hydrological modeling (e.g., Peng et al., 2017; Foster et al., 2020),

our results suggest that the hillslope-based partitioning would be a more effective template for scaling, particularly for mountainous hydrology where aspect and solar radiation play a significant role (Pelletier et al., 2018; Fan et al., 2019).

After the appropriate hillslope scale is defined, we computed the hillslope features (including the hillslope-average of key properties as well as the percent coverage of categorical variables), and applied unsupervised clustering to these features. This clustering process takes advantage of the co-variability between aboveground and belowground properties, and reduces the multidimensional parameter space into one-dimensional classes (Hastie et al., 2013). In the East River watershed, the zones are primarily dictated by elevation, while aspect or radiation poses orthogonal controls to the elevation. Geology and bedrock

are also correlated with elevation, such that hillslopes dominated by crystalline rocks and hydrothermally altered shales and sandstones are primarily found at higher elevations. The plant types are associated with both elevation and radiation, such that the dominant type changes from grassland, aspen, conifer, and grassland to barren from low to high elevations, and conifers are dominant in the north-facing hillslopes.

Instead of relying on a single clustering method, we compared three commonly used clustering methods: hierarchical clustering, k-means, and Gaussian mixture models. Although clustering has been applied in many hydrology or environmental science applications, the selection of methods is often arbitrary and subjective (e.g., Sawicz et al., 2011; Wainwright et al., 2015; Devadoss et al., 2020). Such methodological or conceptual model uncertainties are important to characterize, since they are often larger than parameter uncertainty (Neumann et al., 2003; Ye et al., 2010). Based on the frequency map, we can also

identify the hillslopes that are consistently categorized in each zone as the most representative hillslopes in that zone. In addition, our results show that different methods yield similar zones, provided that the distance metrics are the same. Differences could be explained as the further division of an identified zone, such that one method divided a zone into two, due to a particular metric. For example, the conifer-dominated hillslopes were divided into Zones 3, 4 and 7, which could provide insights into the impact of different bedrock types, and topographic features such as slope.


The identified zones are evaluated with the metrics associated with key watershed functions, including foresummer drought, sensitivity of plant productivity, and river nitrogen export from subcatchments. Drought sensitivity is mainly affected by the dominant plant species in the hillslopes, which is consistent with Wainwright et al. (2020). The conifer-dominated hillslopes have lower sensitivity to the droughts, since they are located mostly in north-facing hillslopes and conifer trees have deeper

roots. However, there is a difference in two grassland-dominated zones: Zone 5 and Zone 1, with the high elevation grassland less sensitive to droughts, possibly owing to increased water availability in higher elevation. In addition, in a separate study by Yan et al. (2021), the soil thickness and associated parameters such as soil diffusion coefficients are found to be distinctly different between the two zones in this domain (Zone 1 and Zone 2).

Nitrogen export is known to be controlled by multiple factors, such as plant retention (Aber et al. 1998), soil cover (Sickman et al., 2002), river corridor features (e.g., floodplains) (Rogers et al. 2021), and geology (Wan et al., 2019; Arora et al. 2020; Maavara et al. 2021). Our analysis showed that the conifer-dominated zone (Zone 3, 4 and 7) is associated with low nitrogen export, while Zone 2, which has the highest fraction of aspen forests, is associated with higher N export. Conifer forests have previously been observed to have high nitrogen retention (Abrahamsen and Stuanes, 1998, Sollins et al., 1981), particularly through ectomycorrhizal uptake (Höberg et al., 2011). By contrast, deciduous aspen forests produce annual leaf litters, with a low C:N content, releasing nitrogen back to the soil (e.g., DeByle et al., 1985; Köchy and Wilson, 1997, Buck & St. Clair, 2012). In addition, Wan et al. (2019) found that shale was a significant geogenic source of nitrogen, based on the intensive observation site located in Zone 2 of this watershed, which is consistent with the global data synthesis (Houlten et al., 2018). We found that the conifer-dominant hillslopes, including those in Zone 3, were in sandstone or igneous rock regions, or in more resistive shale with a lower fracture density. These factors may be combined to reduce the nitrogen delivery to the river in the conifer-dominated hillslopes, and thus the lower observed river nitrogen exports from that subwatershed. We acknowledge that our approach does not explicitly account for microtopography or the small-scale heterogeneity in the wetland areas, which are considered biogeochemical hotspots because they are small in area but have outsized impacts on N export (Duncan et al., 2013; Rogers et al. 2021). However, we assume that zonation can still capture a large-scale pattern such that Zone 1 has the largest coverage of the wetland region.

## 5 Conclusion

In this paper, we have developed a watershed zonation approach for characterizing watershed organization and functions based on the bedrock-to-canopy remote sensing (LiDAR, hyperspectral, and electromagnetic surveys) and spatial data layers (soil, geology, and landcover maps). We first delineated the domain into a set of hillslopes, by computing flow routing based on the LiDAR-based DEM. To choose an appropriate scale of hillslopes, we defined the ratio between the across-hillslope variance (i.e., the variance of hillslope-averaged properties) and the overall variance, and investigated the impact of different threshold drainage areas on the ability of the hillslope-averaged metrics to represent the watershed-scale heterogeneity. We then applied unsupervised clustering to the hillslope features, including the hillslope-averaged metrics of spatial data layers, as well as the percent coverage of categorical variables within each hillslope. Our major findings are: (1) we can define the scale of hillslopes at which the hillslope-averaged metrics can capture the majority of the overall heterogeneity; particularly for elevation and net annual potential radiation; (2) the identified zones (i.e., the groups of hillslopes) have representative characteristics with respect to co-varied bedrock-to-canopy properties; (3) different clustering methods generate similar zonation *patterns*, given the same distance criteria; and (4) the identified zones could provide information about zone-based watershed functions, including foresummer drought sensitivity and river nitrogen exports.

Our approach is similar to catchment clustering and classification of past studies (e.g., Wagener et al. 2007; Krause et al. 2014). Those studies, however, defined the classes of watersheds based on streamflow signatures, which were sparse in space, particularly in headwater catchments. In our study, the expansion of spatial data layers from various remote-sensing data layers provides alternative opportunities to apply clustering to the regions without streamflow measurements. In addition, although our approach is similar the Hydrological Response Unit approach (Flugel, 1997) or hillslope clustering done by Chaney et al. (2018), their main purpose was to parameterize hydrological models. Our analysis confirmed the significance of zonation using the key watershed properties and functions based on observational datasets including novel datasets such as subsurface structures and signatures from airborne EM.

We recognize that every hillslope is unique, each with distinct topographic positions, geology, and vegetation. However, since hillslope-scale studies and experiments are an integral part of watershed science and critical zone science (Hinckley et al., 2014; Tokunaga et al. 2019; Wan et al. 2019), it is important to identify the similarities and differences among the hillslopes. Our approach provides an objective way to classify different hillslopes, including the evaluation of hillslope scales, and the variability associated with different clustering methods. Using our zonation approach, we can guide the experimental plot placement as well as evaluate the representativeness of the selected hillslopes within the domain of interest. After a particular hillslope is identified, we can use pixel-by-pixel clustering to map the heterogeneity within each hillslope associated with microtopography and hillslope positions based on high-resolution images and LiDAR (e.g., Park and Van De Giesen, 2004; Falco et al., 2018; Devadoss et al., 2020; Hermes et al., 2020). This hierarchical representation provides a tractable framework for watershed characterization. Furthermore, Yan et al. (2021) developed two separate parameterizations for the soil evolution model in the two hillslopes that belonged to Zone 1 and 2, and then simulated the variable soil thickness within those hillslopes. In this way, our method can improve model parameterization in large-scale hydrological models by honoring distinct boundaries and water/element export contributions, and provide a new comprehensive way of linking above-and-below-ground properties to watershed functions critical to maintaining water resources.

## Acknowledgments

This material is based upon work supported as part of the Watershed Function Scientific Focus Area funded by the U.S. Department of Energy, Office of Science, Office of Biological and Environmental Research under Award Number DE-AC02-05CH11231. This work was also supported in part by the U.S. Department of Energy, Office of Science, Office of Workforce Development for Teachers and Scientists (WDTS) under the Science Undergraduate Laboratory Internship (SULI) program and Workforce Development and Education at Lawrence Berkeley National Laboratory. The airborne EM data were acquired with funding from the USGS Mineral Resources Program. All data files associated with calculating nitrogen fluxes (gap-filled nitrogen concentrations and gap-filled discharge) are freely available on ESS-DIVE (https://ess-dive.lbl.gov/) for free-public

access. We thank Dr. Lawrence Band at University of Virginia, Dr. Neal Pastick at USGS, and one anonymous reviewer for careful review and constructive comments.

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
