# Peer review of "Watershed zonation approach for tractably quantifying above-andbelowground watershed heterogeneity and functions"

_Hydrology and Earth System Sciences, 2021_

## Referee Comment (RC1)

Review of "Watershed zonation approach for tractably quantifying above-and-belowground watershed heterogeneity and functions"
Author(s): Haruko M. Wainwright et al.
MS No.: hess-2021-228

L. Band, University of Virginia

*General comment:*
This paper describes methods to form multi-dimensional clusters of hillslopes from a set of raster data sets describing watershed properties for an instrumented Rocky Mountain watershed in order to organize and reduce the dimensionality of environmental data. The East River watershed is an alpine glaciated basin, and is an important US Department of Energy (DOE) funded observatory to study hydrologic, biogeochemical, and critical zone processes. The site is characterized by innovative and extensive observations and is used as a test-bed to develop and test a set of earth systems watershed models. As such the paper will be of interest to readers. However, there are a set of clarifications, conceptual, and analytical issues that can be addressed to strengthen the paper.

*Specific comments (scientific questions/issues):*
The study hypothesizes that a set of above and below ground properties co-vary in space, that these covarying properties can be spatially clustered, and have distinct associations with land surface processes and function. These hypotheses are widely accepted and observed, and are the basis of the catena concept. The study is also premised on the assumption that hillslopes provide an organizing template to co-occurring and co-evolved land surface properties, and are therefore a fundamental unit to characterize and simulate the behavior of land surface processes, including the interactions of water, carbon, nutrients and energy. The premise is in agreement with a number of publications over the last few decades, citing the (relatively) closed drainage boundary conditions provided by divides forming fundamental water and solute sources, and lower variation in topographic aspect, and hence, the radiation environment. The hypotheses may need to be restated to demonstrate what new information or concepts are being developed and tested.

The paper investigates whether unsupervised clusters at the hillslope scale generated by three different approaches produce a coherent, organizing template for the multiple spatial variables, and can capture observable variance in two land surface and watershed and behaviors: drought sensitivity and nitrogen export. A single hillslope partition is generated as a template to form clusters, using mean values of the spatial data coverages, without considering within unit variance. More detail should be given to justify the scale of the hillslope partition, as larger or smaller hillslopes may yield different distributions of mean parameter values, and resulting clusters based on altered between and within unit variance. Additional information can be included in a table in the paper or supplement: number of hillslopes, characteristics (e.g. area, relief, etc). It would be useful to inspect the balance of between- and within-unit variance, to demonstrate how much of the total landscape variance is captured by the hillslope partition. The effects of variable hillslope sizes and numbers on the representation of watershed heterogeneity and impacts on coupled water and carbon cycling has previously been investigated in similar Rocky Mountain watersheds (e.g. Band et al., 1991, 1993; and others).

Given the high topographic relief, strong topoclimate gradients in radiation and water balance, and intercorrelation of a number of the spatial datasets used, it is likely that any partitioning of the landscape (hillslopes or grid cells) would produce a reasonable clustering, and may have distinct association with specific landscape functions. The three hypotheses stated in the introduction could be strengthened if the concept of an optimal scale of hillslope partitioning was posed, or included the scale dependence of results. This may require multiple hillslope partitions (different extents of the stream network), and

consideration of subhillslope scale variance – essentially generating multiple realizations of the methods used in the paper across scales.

Soils are often the weak link in distributed watershed data, as discussed by the author. While bedrock geology is used instead of soils in this study, available soil data (SSURGO downloaded from the USDA web soil survey site) shows substantially more spatial detail than the bedrock maps. While SSURGO soils data important to water storage and flow are often highly generalized based on the mapping methods and cartographic presentation, there have been a number of methods published over the last decades to develop estimates of soil properties at resolutions comparable to available terrain information (e.g. Zhu et al, 1997), and more recently Chaney et al (2016, 2019) published a 30m soil property dataset for CONUS. The authors should better outline why soils, a central critical zone component, were not used as part of the clustering. Similarly, while aspect was discussed in the paper as a central influence on critical zone behavior, it was not included. While potential radiation may explain much of the information aspect may convey, aspect is a simpler and more widely available measure (but needs to be treated as a circular variable or transformed into a linear surrogate, such as the widely used "southness").

The goals of the clustering are an important driver of the methods. Much of the hydrologic and biogeochemical behavior of watersheds is based on sub-hillslope processes. As an example, the role of riparian areas in modulating both runoff and nutrient export has been heavily cited (e.g. McGlynn and McDonnell, 2003a,b). The last line of the paper suggests the zonation methods presented can guide experimental plot placement to better quantify and understand water/element export contributions. Plots are subhillslope scales, and position within the internal flow structure of the hillslope is a critical control. This is a major tenet of the critical zone approach.

*Technical corrections, clarifications:*
1. P.7, paragraph 185: it is not clear how the clusters were aligned or compared between the clustering methods. I presume the unsupervised clusters are developed independently between methods. Are you renumbering according to similarity?
2. Spell out acronyms the first time used (e.g. NDVI, NDWI, etc) even if these are well known by some communities
3. Figure 1: Clarify the position of the subcatchments. It is given as ordered from right to left (better to state east to west), but Slate River and Coal Creek cannot be distinguished as they appear to be equally "left" or west. A simple label would obviate this.
4. Figure 4: It is not clear if these plots are aggregated over all clustering methods or are for one.
5. P.11, paragraph 255: Incomplete sentence "The Rock and Gothic subcatchments, which are predominantly within conifer dominated Zone 3 (Figure b)." Figure 5b?
6. Figure 6d,e: y-axis is given as MG/m$^3$. Is this a concentration or a mass export? Typo in units? Perhaps MG/km$^2$? Figure 6e is unconvincing as there are two widely separated clusters of points that are interpreted as a trend.

Citations:

L.E. Band, D.L. Peterson, S.W. Running, J.C. Coughlan, R. Lammers, J.Dungan and R.R. Nemani, 1991. "Ecosystem processes at the watershed level: Basis for distributed simulation," Ecological Modeling, 1991, v.56, p.171- 196.

L.E. Band, P. Patterson, R.R. Nemani and S.W. Running, 1993. "Forest ecosystem processes at the watershed scale: 2. Adding hillslope hydrology." Agricultural and Forest Meteorology, 1993, v.63, p.93-126.

Chaney, N. W., Minasny, B.,Herman, J. D., Nauman, T. W., Brungard, C., Morgan, C. L. S., et al. (2019). POLARIS soil properties: 30-m probabilistic maps of soil properties over the contiguous United States. *Water Resources Research*, *55*, 2916–2938. https://doi.org/10.1029/2018WR022797

NW Chaney, EF Wood, AB McBratney, JW Hempel, TW Nauman, … 2016. POLARIS: A 30-meter probabilistic soil series map of the contiguous United States.  Geoderma 274, 54-67

BL McGlynn, JJ McDonnell, 2003. Quantifying the relative contributions of riparian and hillslope zones to catchment runoff.  Water Resources Research 39 (11)

BL McGlynn, JJ McDonnell, 2003. Role of discrete landscape units in controlling catchment dissolved organic carbon dynamics. Water Resources Research 39 (4)

A. Zhu, L.E. Band, R. Vertessy and B. Dutton, 1997. Soil property derivation using a soil land inference model (SoLIM). Soil Science Society America Journal, v.61(2), p.523-533.

---

## Author Comment (AC1)

Review of "Watershed zonation approach for tractably quantifying above-and-belowground watershed heterogeneity and functions"

Author(s): Haruko M. Wainwright et al.

MS No.: hess-2021-228

L. Band, University of Virginia

General comment:

This paper describes methods to form multi-dimensional clusters of hillslopes from a set of raster data sets describing watershed properties for an instrumented Rocky Mountain watershed in order to organize and reduce the dimensionality of environmental data. The East River watershed is an alpine glaciated basin, and is an important US Department of Energy (DOE) funded observatory to study hydrologic, biogeochemical, and critical zone processes. The site is characterized by innovative and extensive observations and is used as a test-bed to develop and test a set of earth systems watershed models. As such the paper will be of interest to readers. However, there are a set of clarifications, conceptual, and analytical issues that can be addressed to strengthen the paper.

> Answer: We appreciate Dr. Band's insightful and constructive comments. The suggested papers (Band, 1989; Band et al., 1991; Band et al., 1993) were very insightful to consider the scale of hillslopes. We have included additional results and revised the manuscript accordingly. The detailed explanations were below.

Specific comments (scientific questions/issues):

The study hypothesizes that a set of above and below ground properties co-vary in space, that these covarying properties can be spatially clustered, and have distinct associations with land surface processes and function. These hypotheses are widely accepted and observed, and are the basis of the catena concept. The study is also premised on the assumption that hillslopes provide an organizing template to co-occurring and co-evolved land surface properties, and are therefore a fundamental unit to characterize and simulate the behavior of land surface processes, including the interactions of water, carbon, nutrients, and energy. The premise is in agreement with a number of publications over the last few decades, citing the (relatively) closed drainage boundary conditions provided by divides forming fundamental water and solute sources, and lower variation in topographic aspect, and hence, the radiation environment. The hypotheses may need to be restated to demonstrate what new information or concepts are being developed and tested.

> Answer: We agree that the co-evolution theory has been established across the literature. Our unique contributions are that: (1) this paper has established a clustering approach to capture co-varied bedrock-to-canopy properties by a set of

zones, which is equivalent to reducing a multidimensional parameter space to one-dimensional representation and makes characterization more tractable, and (2) this is the first paper that have a quantitative metric of bedrock properties at the watershed scale based on airborne electromagnetic survey (AEM), showing the co-variability of bedrock resistivity, elevation and other properties. We have emphasized these points in the conclusion section.

The paper investigates whether unsupervised clusters at the hillslope scale generated by three different approaches produce a coherent, organizing template for the multiple spatial variables, and can capture observable variance in two land surface and watershed and behaviors: drought sensitivity and nitrogen export.  A single hillslope partition is generated as a template to form clusters, using mean values of the spatial data coverages, without considering within unit variance.  More detail should be given to justify the scale of the hillslope partition, as larger or smaller hillslopes may yield different distributions of mean parameter values, and resulting clusters based on altered between and within unit variance. Additional information can be included in a table in the paper or supplement: number of hillslopes, characteristics (e.g. area, relief, etc).  It would be useful to inspect the balance of between- and within-unit variance, to demonstrate how much of the total landscape variance is captured by the hillslope partition.  The effects of variable hillslope sizes and numbers on the representation of watershed heterogeneity and impacts on coupled water and carbon cycling has previously been investigated in similar Rocky Mountain watersheds (e.g. Band et al., 1991, 1993; and others).

Answer: Thank you for this great suggestion and these papers. We agree that different sets of hillslopes can be defined depending on the scale or the threshold flow accumulation (or drainage) area to define stream segments. In the original manuscript, we have set the threshold to match the observed streams. In the revised manuscript, following Band (1989) and Band et al (1991), we have generated different sets of hillslopes based on the flow accumulation threshold to define streams.

For the hillslope metrics that are defined by the average (e.g., average elevation, average slope), we evaluated the within-hillslope variability and across-hillslope variability of each property, compared to the overall variability. We consider that the variance of pixel-by-pixel properties represents the overall variability over this domain, while the variance of hillslope-averaged metrics represents the variability across the hillslopes. We computed the ratio between the across-hillslope variance and overall variance, representing how well the hillslope-averaged metrics can capture watershed-scale variability.

Figure X1 (below; Figure 2 in the manuscript) shows this ratio as a function of the threshold drainage area. The ratio is high for the elevation, radiation, snow, and bedrock resistivity up to 0.75, which means that the hillslope-averaged metrics

capture the watershed-scale variability of these variables and that the within-hillslope variability is small compared to the across-hillslope variability. TWI and NDVI, on the other hand, have a low ratio, which means that the within-hillslope variability is significant. The ratio increases as the drainage area decreases, since the smaller the hillslopes are, the better they capture the small-scale variability. However, the variance ratio of the elevation and radiation reaches a plateau with the drainage area around $10^6$m^2. Based on this result, we selected 810,000 $m^2$ as the threshold drainage area in the subsequent analysis.

In addition, as a contrast, we included the variance of each property at the upscaled pixel (i.e., taking the averaged value in larger pixel sizes compared to the original 9-meter pixel) compared to the overall variance (Figure X1b). The ratio between the across-upscaled-pixel variance and the overall variance increases as the pixel size decreases. Different from the hillslope average, the ratio for radiation keeps increasing without reaching the plateau. This means that a representative size or scale of pixels does not exist when we use pixel-based upscaling.

We also considered the implication of this plateau. Hillslopes have been used as an organizational unit for watershed hydrology (Fan et al., 2019) as well as critical zone science (Brantley et al., 2019). In particular, Pelletier et al. (2019) showed a significant impact of hillslope aspects and slopes on evapotranspiration, weathering, nutrient cycling, and others. The plateau of the ratio between the across-hillslope and overall variance suggests that there is a representative hillslope scale for radiation (which is a function of slope and aspect), at which the hillslope average can capture the majority of the pixel-by-pixel variability at the watershed scale and the within-hillslope variability is small. At the same time, the ratio for NDVI (associated with plant dynamics) and TWI (associated with soil moisture) continues to increase as the hillslopes become smaller. This is due to the fact that they are significantly affected by microtopography and within-hillslope positions (i.e., toe slopes). These results suggest that we may define a hierarchical representation of spatial heterogeneity such that the watershed-scale variability of elevation and radiation is captured by hillslopes, while the other properties are defined within each hillslope.

We included these texts and Figure 2 in the manuscript (the third paragraph in Section 3.1 in the methodology section; Section 4.1 in the result section; the third paragraph in the discussion section)

[Figure]

Figure X1 (Figure 2 in the manuscript). Variance ratio (a) between the across-hillslope variance (i.e., the variance of hillslope-averaged metrics across all the hillslopes) and the overall variance (i.e., the variance of the pixel-by-pixel properties) as a function of the threshold drainage area, and (b) between the across-upscaled-pixel variance and the overall variance as a function of the pixel area.

Given the high topographic relief, strong topoclimate gradients in radiation and water balance, and intercorrelation of a number of the spatial datasets used, it is likely that any partitioning of the landscape (hillslopes or grid cells) would produce a reasonable clustering and may have distinct association with specific landscape functions.  The three hypotheses stated in the introduction could be strengthened if the concept of an optimal scale of hillslope partitioning was posed, or included the scale dependence of results.  This may require multiple hillslope partitions (different extents of the stream network), and consideration of subhillslope scale variance – essentially generating multiple realizations of the methods used in the paper across scales.

> Answer: Thank you very much for the suggestion. As described above, we have created multiple scales of hillslopes, and evaluated the variance within/across hillslopes.

Soils are often the weak link in distributed watershed data, as discussed by the author. While bedrock geology is used instead of soils in this study, available soil data (SSURGO downloaded from the USDA web soil survey site) shows substantially more spatial detail than the bedrock maps.  While SSURGO soils data important to water storage and flow are often highly generalized based on the mapping methods and cartographic presentation, there have been a number of methods published over the last decades to develop estimates of soil properties at resolutions comparable to available terrain information (e.g. Zhu et al, 1997), and more recently Chaney et al (2016, 2019) published a 30m soil property dataset for CONUS.  The authors should better outline why soils, a central critical

zone component, were not used as part of the clustering. Similarly, while aspect was discussed in the paper as a central influence on critical zone behavior, it was not included. While potential radiation may explain much of the information aspect may convey, aspect is a simpler and more widely available measure (but needs to be treated as a circular variable or transformed into a linear surrogate, such as the widely used "southness").

Answer: We have agreed with this comment, and included soil texture (%clay, and %sand) from Chaney et al (2016). Figure X2 (below; Figure 2 in the manuscript) shows that soil texture is correlated with the elevation, slope, and other metrics at the hillslope scale. We have updated the clustering results as well. Although the zones are slightly different from earlier results, the general observations (e.g., types of hillslopes, validation) are the same, since %sand and %clay are correlated with elevation and other metrics

[Figure]

Figure X2. Correlation and scatter plots (Pearson's correlation coefficients) among selected hillslope features (Table S1). The * sign represents p-values < 0.01.

In terms of the aspect, we capture the effect of the aspect by the net potential annual radiation, which is computed as a function of slope, aspect, and solar angle at each

pixel. It is a more representative metric for the magnitude of solar radiation annually (without considering the cloud effects) than the aspect. We have included this clarification in the first paragraph of Section 3.1 as "The annual net potential solar radiation – a function of the aspect, slope and solar angle – is considered as a better metric to represent the intensity of solar radiation than the aspect itself which is circular (0 and 360 degrees are the same; Wainwright et al., 2020)."

The goals of the clustering are an important driver of the methods. Much of the hydrologic and biogeochemical behavior of watersheds is based on sub-hillslope processes. As an example, the role of riparian areas in modulating both runoff and nutrient export has been heavily cited (e.g. McGlynn and McDonnell, 2003a,b). The last line of the paper suggests the zonation methods presented can guide experimental plot placement to better quantify and understand water/element export contributions. Plots are subhillslope scales, and position within the internal flow structure of the hillslope is a critical control. This is a major tenet of the critical zone approach.

Answer: We agree with this comment. Our group indeed has extensive experience mapping pixel-by-pixel heterogeneity within a hillslope based on high resolution images (e.g., Falco et al., 2018; Devadoss et al., 2020; Hermes et al., 2020). We have added this sentence in the last paragraph of the conclusion section "After a particular hillslope is identified, we can use pixel-by-pixel clustering to map the heterogeneity within each hillslope associated with microtopography and hillslope positions based on high-resolution images and LiDAR (e.g., Park and Van De Giesen, 2004; Falco et al., 2018; Devadoss et al., 2020; Hermes et al., 2020). This hierarchical representation provides a tractable framework for watershed characterization."

Technical corrections, clarifications:

1. 7, paragraph 185: it is not clear how the clusters were aligned or compared between the clustering methods. I presume the unsupervised clusters are developed independently between methods. Are you renumbering according to similarity?

   Answer: Yes, we evaluated the frequency maps (Figure S4), which show how frequent each hillslope is categorized into a particular zone among three methods. We also included the explanation to compare the zones from the three methods, "To compare the results from these three methods, we first identified six common zones across the three methods that have the overlapping coverage, starting from the GMM-based map as a basis."

2. Spell out acronyms the first time used (e.g. NDVI, NDWI, etc) even if these are well known by some communities.

   Answer: Yes, we have added the full names in the second paragraph of Section 3.1.

3. Figure 1: Clarify the position of the subcatchments. It is given as ordered from right to left (better to state east to west), but Slate River and Coal Creek cannot be distinguished as they appear to be equally "left" or west. A simple label would obviate this.

Answer: Yes, we have assigned the indices to the watersheds, and included them in Figure 1a, as (A) East River, (B) Washington Gulch, (C) Slate River and (D) Coal Creek.

4. Figure 4: It is not clear if these plots are aggregated over all clustering methods or are for one.

Answer: Yes, it is aggregated. We have included the explanation in the third paragraph in Section 4.2., "In addition, the statistics are computed by aggregating the zonation from the three methods (Figure S4) such that the zonation of each method has the weight of 1/3."

5. 11, paragraph 255: Incomplete sentence "The Rock and Gothic subcatchments, which are predominantly within conifer dominated Zone 3 (Figure b)." Figure 5b?

Answer: Thank you for pointing this out. We revised this sentence as "The Rock and Gothic sub-catchments, which are predominantly within conifer-dominated Zone 3, 4 and 7 (Figure 7b), have a lower nitrogen expected from this simple scaling relationship"

6. Figure 6d,e: y-axis is given as MG/m$_3$. Is this a concentration or a mass export? Typo in units? Perhaps MG/km$_2$? Figure 6e is unconvincing as there are two widely separated clusters of points that are interpreted as a trend.

Answer: This is the normalized value of dividing the N mass export (MG/year) by the annual discharge (M^3/year). We believe that the unit is accurate. We added "mass export" (instead of just export) to clarify this point.

We changed Figure 6d and e to the boxplots in Figure 7d and e, so that the statistical test is performed as two groups (such as high or low coverages of each zone) instead of a trend.

Citations:

L.E. Band, D.L. Peterson, S.W. Running, J.C. Coughlan, R. Lammers, J.Dungan and R.R. Nemani, 1991. "Ecosystem processes at the watershed level: Basis for distributed simulation," Ecological Modeling, 1991, v.56, p.171- 196.

L.E. Band, P. Patterson, R.R. Nemani and S.W. Running, 1993. "Forest ecosystem processes at the watershed scale: 2. Adding hillslope hydrology." Agricultural and Forest Meteorology, 1993, v.63, p.93-126.

Chaney, N. W., Minasny, B.,Herman, J. D., Nauman, T. W., Brungard, C., Morgan, C. L. S., et al. (2019). POLARIS soil properties: 30-m probabilistic maps of soil properties over the contiguous United States. Water Resources Research, 55, 2916–2938. https://doi.org/10.1029/2018WR022797

NW Chaney, EF Wood, AB McBratney, JW Hempel, TW Nauman, … 2016. POLARIS: A 30-meter probabilistic soil series map of the contiguous United States. Geoderma 274, 54-67

BL McGlynn, JJ McDonnell, 2003. Quantifying the relative contributions of riparian and hillslope zones to catchment runoff. Water Resources Research 39 (11)

BL McGlynn, JJ McDonnell, 2003. Role of discrete landscape units in controlling catchment dissolved organic carbon dynamics. Water Resources Research 39 (4)

Zhu, L.E. Band, R. Vertessy and B. Dutton, 1997. Soil property derivation using a soil land inference model (SoLIM). Soil Science Society America Journal, v.61(2), p.523-533.

---

## Author Comment (AC2)

The paper titled "Watershed zonation approach for tractably quantifying above-and-belowground watershed heterogeneity and functions" develops a watershed zonation approach for characterizing watershed organization and function. The authors use multiple high resolution spatial datasets available over the East River watershed to explore this relationship at the hillslope level. Their use of process-driven observations (Annual N export) is an important contribution as it shows that the zonation is indeed connected to the local processes. I recommend that this paper be accepted for publication after major revisions.

Thank you very much for the detailed review and constructive comments. We have revised the manuscript accordingly.

1. The steps used to delineate the watersheds is clear in the current manuscript. However, the step to delineate the hillslopes remains unclear. I am assuming it involves just splitting each watershed into a left/right hillslopes and a "headwaters" hillslope if it exists; however, this needs to be made more explicit. On a related note, the title is misleading since the clustering is performed at the hillslope level. I would suggest that the title be clarified so that it is clear that it is hillslope-level and not simply watershed-level.

Answer: We have expanded the explanation of hillslope delineation in the first paragraph of Section 3.1. We also changed the title to "Watershed zonation through hillslope clustering for tractably quantifying above-and-belowground watershed heterogeneity and functions"

2. "This is consistent with Wood et al. (2011), concluding that the order of 100 m is a sufficient resolution for representing hydrological fluxes" is a stretch. There is a lot more heterogeneity that will matter at scales finer than hillslopes even if it is not captured in your data; furthermore, the role of hillslope hydrology is variable depending on the topographic environment. This sentence is not necessary to show your point; I would remove it.

Answer: We agreed and removed this sentence.

3. Line 180 - The reason why the hillslope-level correlations is better than pixel-to-pixel can be deceptive. This is probably a combination of the processes being more connected at 100 meter scales but also just simply because you are removing random noise from the higher resolution data by aggregating at the hillslope level. I think it would be useful to test how the correlations vary as you upscale the original regular grid maps (e.g., 50 m, 100 m, 250 m...). You would do a pixel-to-pixel comparison for these as well. It would be a strong result if the hillslope approach still wins out. My hypothesis is that it won't and that they will be pretty close. This analysis would be useful within the paper or could be placed in the supplement.

Answer: Thank you for this great suggestion. We indeed confirmed that upscaling the pixels improves the correlations among metrics. We included the pixel-by-pixel correlations at different pixel sizes in the supplementary material (Figure S2).

However, we still think that the hillslopes are still more effective units for capturing the watershed heterogeneity than pixel-based upscaling. In response to Dr. Band's comments, we have computed the across-hillslope variance (i.e., the variance of the hillslope averaged metrics) compared to the overall pixel-by-pixel variance (Figure X1a). In addition, we computed the variance of the upscaled pixel metrics compared to the overall variance (Figure X1b). The ratio of the across-hillslope variance or upscaled-pixel variance decreases as the threshold drainage area or the pixel area increases, since the averaged values do not account for the overall variability fully. However, in the hillslope averaging (Figure X1a), there is a plateau of the area with some metrics like elevation, radiation and peak SWE (which are known to be critical for key watershed processes). This means that there is a certain hillslope size, up to which the variability within each hillslope is limited and the hillslope-averaged metrics can capture the overall variance. Such a plateau does not exist for the case with upscaled pixels, since larger pixels can contain different aspects of hillslopes within. This shows that the hillslope averaging is more effective than averaging within upscaled pixels, which is consistent with Band et al. (1991).

[Figure]

Figure X1 (Figure 2 in the manuscript). Variance ratio (a) between the across-hillslope variance (i.e., the variance of hillslope-averaged metrics across all the hillslopes) and the overall variance (i.e., the variance of the pixel-by-pixel properties) as a function of threshold drainage area, and (b) between the across-upscaled-pixel variance and the overall variance as a function of the pixel area.

4. The current discussion has a number of disjointed paragraphs and ideas. I would encourage to split it up into discussion and conclusion sections and then to subdivide the discussion section into subsections.

Answer: We have improved the connectivity between paragraphs in the discussion section, and created a conclusion section in the revised manuscript.

5. Figure 1 - The legend of NLCD is not comprehensive. There are developed/urban areas on the map but they are not referenced in the legend.

Answer: Thank you for pointing this out. We have revised the legend in Figure 1b.

6. Figure 3 - The 10 on the colorbar is cut off

Answer: Thank you for pointing this out. We have changed Figure 4 (formerly Figure 3)

Citation: https://doi.org/10.5194/hess-2021-228-RC2

---

## Referee Report (RR1)

Review of revised manuscript, "Watershed zonation approach for tractably quantifying above-and-belowground watershed heterogeneity and functions"
Author(s): Haruko M. Wainwright et al.
MS No.: hess-2021-228

The authors have carried out significantly more analysis following my suggestions of investigating multiple scales of hillslope partitioning, and the scale dependence of hillslope characterization of covarying environmental factors. Results are interesting, with an emergent scale that appears to provide improved partitioning of a set of environmental drivers, reflecting the organizing framework of topoclimatic and geologic patterns. Interestingly, the identified scale is very close to the scale developed for the Representative Elementary Area analysis for the Coweeta watershed in North Carolina by Wood et al. 1988[1]. More detailed soil information is also included with the adoption of Chaney's POLARIS dataset.

The only substantive suggestion I have is that these new developments should be reflected in the major hypotheses and conclusions of the paper. As previously pointed out, hypothesis 1: suites of aboveground/belowground properties co-vary with each other, is a widely accepted, broadly known and demonstrated tenet of a number of environmental sciences, and has been widely published on and quantified. What this paper does more uniquely is demonstrate the scale (hillslope) dependence of this covariation, and sensitivity to partitioning method (e.g. hillslopes vs. upscaled grids). I think that making this clear up front in the stated hypotheses would better reflect the revised major analytical methods and conclusions.

Otherwise, with minor corrections, I think the paper is ready to go. Most of these minor corrections are spelling, typos, etc. in the revised sections that a good read will find. The authors have done a good a job with the additional analysis – I look forward to seeing additional research follow-on.

[1] https://doi.org/10.1016/0022-1694(88)90090-X

---

## Author Response (AR2)

Responses to the comments by Dr. Band

*The authors have carried out significantly more analysis following my suggestions of investigating multiple scales of hillslope partitioning, and the scale dependence of hillslope characterization of covarying environmental factors. Results are interesting, with an emergent scale that appears to provide improved partitioning of a set of environmental drivers, reflecting the organizing framework of topoclimatic and geologic patterns. Interestingly, the identified scale is very close to the scale developed for the Representative Elementary Area analysis for the Coweeta watershed in North Carolina by Wood et al. 1988[1]. More detailed soil information is also included with the adoption of Chaney's POLARIS dataset.*

> Response: We appreciate your constructive comments that have improved this manuscript. Thank you for suggesting another paper on the scale of hillslopes. It is indeed fascinating to see a similar scale that emerged in two different sites. We have included this paper in the discussion section.

*The only substantive suggestion I have is that these new developments should be reflected in the major hypotheses and conclusions of the paper. As previously pointed out, hypothesis 1: suites of aboveground/belowground properties co-vary with each other, is a widely accepted, broadly known and demonstrated tenet of a number of environmental sciences, and has been widely published on and quantified. What this paper does more uniquely is demonstrate the scale (hillslope) dependence of this covariation, and sensitivity to partitioning method (e.g. hillslopes vs. upscaled grids). I think that making this clear up front in the stated hypotheses would better reflect the revised major analytical methods and conclusions.*

> Response: we have revised the research hypothesis to emphasize the importance of hillslopes to capture the co-varied properties: "We hypothesize that (1) a hillslope is an appropriate unit to capture the watershed-scale heterogeneity of key bedrock-through-canopy properties, and to quantify the co-variability of these properties representing coupled ecohydrological and biogeochemical interactions, (2) we can identify a group of hillslopes or watershed-scale zones ¬that have unique distributions of these properties relative to neighboring parcels, and (3) the identified zones can capture the variability of key watershed functions"

*Otherwise, with minor corrections, I think the paper is ready to go. Most of these minor corrections are spelling, typos, etc. in the revised sections that a good read will find. The authors have done a good a job with the additional analysis – I look forward to seeing additional research follow-on.*

> Response: Thank you for pointing this out. A technical editor has reviewed the manuscript, and we have fixed grammatical errors.